# Nanocomposites as Substituent of Cement: Structure and Mechanical Properties

**DOI:** 10.3390/ma16062398

**Published:** 2023-03-16

**Authors:** Roxana Rada, Daniela Lucia Manea, Ramona Chelcea, Simona Rada

**Affiliations:** 1Department of Civil Engineering and Management, Faculty of Civil Engineering, Technical University of Cluj-Napoca, 400114 Cluj-Napoca, Romania; 2Department of Physics and Chemistry, Faculty of Materials and Environmental Engineering, Technical University of Cluj-Napoca, 400114 Cluj-Napoca, Romania; 3National Institute of Research and Development for Isotopic and Molecular Technologies, 400293 Cluj-Napoca, Romania

**Keywords:** composite-cement, XRD, IR, NMR, Vickers hardness

## Abstract

To date, the scientific research in the field of recycling of construction and demolition wastes was focused on the production of concrete, cements, and bricks. The attainment of these products was limited to the addition of suitable binder contents, such as lime or cement, compaction, and possibly heat treatment, without a concrete recycling method. In this paper, new cement materials consisting of 2.5 weight% composite and originating from construction and demolition waste powder, were prepared and investigated in view of applications in the construction industry as a substituent of cement. The materials with recycled powder from construction and demolition wastes were characterized by X-ray diffraction (XRD), infrared (IR) and nuclear magnetic resonance (NMR) spectroscopy. The XRD data indicate vitroceramic structures with varied crystalline phases. The NMR relaxometry data show four reservoirs of water associated with bounded water and with three types of pores in the composite construction material. The micro-Vickers hardness was measured to reflect the influence of composite nature in the local mechanical properties of the composite-cement for the mixture with Portland cement and (EC) expired cement.

## 1. Introduction

Large quantities of waste are produced during the construction of developments, when buildings are refurbished and demolished at the end of their lives. The management of construction and demolition (C&D) wastes results in considerable environmental impacts. The use of alternative management routes can generate environmental improvements and cost savings. In the European Union, construction and demolition wastes represent one third of all solid municipal waste. Construction and demolition wastes contain a wide variety of materials, such as concrete, bricks, wood, glass, metal, and plastic [1].

The main scientific research was focused on the recycling of construction of concrete, cement, and bricks [2,3]. Typically, the main objectives of these studies involve the possibility of obtaining new products using lime or cement as binders without the involvement of a recycling technology [4].

The recycling of secondary materials is environmentally beneficial, since in the primary production of materials, significant amounts of raw materials and energy are used. In this sense, the separation of metals from construction and demolition wastes and the recycling of other metal products represent a priority for C&D waste management. Moreover, there are considerable financial benefits, which already drive the recycling of many materials. In this paper, two types of construction and demolition wastes, namely, broken glasses and metals (iron, cash iron, and lead) were tested for recycling.

Wood waste is the second largest component of construction and demolition after concrete. The amount of wood is around 10% of all materials deposited in landfills. The legislation classifies timber and wood products as a C&D waste component with limited reuse if the material is contaminated by environmentally harmful materials. Wood waste from construction and demolition activities is usually delivered for the processing of boiler fuels and pellets. In this paper, reutilization of ash from the remaining residue after the combustion of wood waste in the construction and demolition sector is the topic of the research.

Concrete is widely used as a construction material and requires theconsumption of a significant amount of cement since it is the main binding material. The production of Portland cement has a greenhouse effect due to the emission of CO_2_ into the atmosphere. The production of 1 ton of Portland cement results in 1 ton of CO_2_ [5]. The Hardened cement paste is the most important component of concrete, which is a porous material with pore sizes varying from several micrometers to nanometers. The performance of concrete is derived from the precipitation of calcium silicate hydrate (C-S-H) [6]. A detailed understanding of the C-S-H structure in the cement paste and the mechanisms determining the concrete properties can directly contribute to the increase in the durability of concrete materials and the reduction in the CO_2_ emissions for combating climate change by recycling.

The possibility of introducing the recycled concrete powder into concrete as a substitute for cement was studied in the last few years [7,8,9]. The addition of 5 wt% silica fume and 20 wt% fly in the cement shows that the abrasion resistance and mechanical property were increased by about 4–9% [7].

The concept of incorporating nanomaterials in the cement is new and extensively exploited in the construction industry [10]. To date, the incorporation of various nanomaterials, including nano-SiO_2_, nano-Al_2_O_3_, nano-TiO_2_, carbon nanotubes, grapheme, and iron oxide nanoparticles into cementitious materials were demonstrated to accelerate cement hydration and improve mechanical strength [11,12,13,14,15].

Metallic oxides originally found in cement, such as free CaO (the main accessible oxide), SiO_2_, Fe_2_O_3_, FeO and Al_2_O_3_ can easily react in the long term with the incorporated nanomaterial [12].

Briefly, during the recent years, many studies were focused on the effect of nanoparticles in the construction materials, especially hardened cement paste, cement mortar, and concrete. The literature data suggest that a lower content of nanoparticles can improve the mechanical properties and durability of high performance concrete [16]. Despite progress, the research on nanoparticles-cement is still inadequate [17].

The goal of sustainable development of the built environment is to minimize the consumption of natural resources and reduce the consumption of cement. In this paper, the composite materials are preparedas raw materials using the recycled powders from construction and demolition wastes, such as broken glasses, cash iron, iron, lead, or ash powder. Metallic powders or waste ash will be incorporated into the glass network; therefore, the main nanoparticles are enclosed in the glass network and the metallic oxides of cement are not accessible to produce new crystalline phases. Vickers hardness was used as an indicator of the changes in the local mechanical properties of the composite-cement materials. Our aim was to gain a more detailed understanding of the structural and mechanical properties of the composite-cement materials using XRD, IR and NMR data and the distribution of Vickers hardness. Identification of the different pore types on the surface measured by ^1^H NMR relaxometry was also addressed in this paper.

## 2. Experimental Procedure

Chemicals used in this paper were waste glasses collected from broken window, waste powder (recycled powder from construction and demolition wastes, such as cash iron, iron, lead, or ash powder), NaOH, HCl, gray Portland cement, and expired Portland cement. Waste cash iron and iron powder were dissolved in 10% H_2_SO_4_ solution.

Waste glass powder was added to 1N NaOH solution for partial dissolving in a porcelain capsule placed on a mechanical stirrer. When the glass content was dissolved with 1N HCl solution, a stoechiometric amount of waste powder was added to the solution. The experiment was performed with constant stirring speed at 50 °C for 15 min, and continued at 100 °C for 5 min. Finally, the temperature was increased to 250 °C and the composite product was obtained.

Composite-cement materials were prepared with gray Portland cement and, in every mixture, 2.5 weight% of cement amount was substituted by the composite product. The composition of the cement-based materials with recycled powder from construction and demolition wastes is listed in Table 1. The water to cement ratio was 0.3:1 for the validated cement and 0.4:1 for the expired cement, respectively.

The samples were characterized by X-ray diffraction using a Smart Lab Rigaku diffractometer, with a monochromator of graphite for the Cu-Kα radiation (λ = 1.54 Å) at room temperature. The data were collected in the 2θ range 10–60° with scanning step size of 0.01° and step time of 0.1 s per step. For phase identification, the material was carried out using the Match! software. The PDF-2 database was used for the identification of crystalline phases.

The Fourier Transform InfraRed (FTIR) spectra of the glasses were obtained in the 350–2000 cm^−1^ spectral range with a JASCO FTIR 6200 spectrometer (JASCO, Tokyo, Japan) using the standard KBr pellet disc technique. The spectra were carried out with a standard resolution of 2 cm^−1^.

The samples were subjected to nuclear magnetic resonance (NMR) measurements using a low field Bruker Minispec NMR spectrometer (Bruker, Billerica, MA, USA) operating at 19.69 MHz proton frequency.

A Nova microdurimeter equipped with the microscope was used to measure the micro-Vickers hardness. A penetrator with diameter D was operated with a load F of 0.3 kgF for an interval of 15 s. After unloading the penetrator, the diagonals of the trace of pyramidal contour were measured and the values of HV hardness (expressed in MPa) were determined with the device software.

## 3. Results and Discussion

### 3.1. X-ray Diffraction Analysis of the Composite Materials

X-ray diffractograms of the broken glasses and prepared composite materials are shown in Figure 1. The XRD pattern reveals two halos that are characteristic of the amorphous structure of the waste glass powder. The analysis of XRD data proves that the presence of different crystalline phases depends on the doping nature.

For the glassdoped with cash iron (denoted as C), the formation of FeCl_2_·(H_2_O)_2_ with a monoclinic structure was detected. With the introduction of iron content in the glass network, the amount of FeCl_2_(H_2_O)_2_ crystalline phase decreases and the presence of Fe_3_O_4_ crystalline phase with a cubic structure was evidenced.

In the lead composite denoted with L, the main phase of CaPbO_3_ crystalline phase with an orthorhombic structure was accompanied by the secondary phase of PbCl_2_ crystalline phase.

By doping with ash, two phases of CaCO_3_ and KCl with the most intense lines were noted in the XRD data.

The average crystallite size of particle D was determined by the Debye Scherrer equation [18], considering that λ is the X-ray wavelength (0.154 nm), β is the broadening of the diffraction peak in radians (full width at half maximum of the peak), and θ is the diffraction angle for maximum peak in radians. The average crystallite size of prepared composite particles for different dopant types is extracted using the Debye Scherrer equation and is shown in Table 2.

The particle size in the prepared composites increases with the doping of lead and ash. The values of obtained particle size in the composite for the high intensity peak confirm the nanostructure properties. Literature data on the particle size effect of recycled brick powder from the blended cement paste show that the reduction in particle sizes improves the pore structure [19]. The higher particle sizes in the samples doped with lead and ash can have a significant impact on porosity and, at later stages, transport the properties of cement structure.

### 3.2. FTIR Spectra of the Composite Materials

Infrared spectra of broken glasses and prepared composite materials are presented in Figure 2. The analysis of IR data indicates specific domains of silicate units (800 and 1300 cm^−1^) and structural units of the metallic ions (370 and 550 cm^−1^). The first region of IR spectra situated between 370 and 550 cm^−1^ can be attributed to the stretching vibrations of metal (Me)—oxygen bonds in the [MeO_n_] structural units, where n = 4 and 6 overlapped with the stretching vibrations in the silicate glass network [20,21]. For sample I, this region corresponds to the stretching vibrations of the Fe-O bonds with iron cations located in octahedral and tetrahedral sites, respectively and the formation of Fe_3_O_4_ crystalline phase, in accordance with XRD data. For sample L, the intensity of the IR band centered at about 460 cm^−1^ corresponds to the stretching vibrations of Pb-O bonds in the [PbO_4_] structural units.

The high intensity bands region located between 800 and 1300 cm^−1^ is assigned to the stretching vibrations of the Si-O bonds in varied silicate structural units. By doping, this region differs from features of the host matrix since it appears as new shoulders and characteristic IR bands, which indicates a drastic depolymerization of silicate network. These IR bands were assigned to symmetric Si-O stretching vibrations of silicate units containing [SiO_4_] tetrahedral units with zero, one, two, three, and four non-bridging oxygens, namely, tectosilicate, disilicate, metasilicate, pyrosilicate, and orthosilicate units denoted as Q4, Q3, Q2, Q1, and Q0, respectively [22]. These IR features allow for thedistinguishment of bands located at about 1100–1250 cm^−1^, 1000–1100 cm^−1^, 900–1000 cm^−1^, near 900 cm^−1^, and near 850 cm^−1^ of each dominant at the tectosilicate, disilicate, metasilicate, pyrosilicate, and orthosilicate units.

The IR bands are characterized by a broad asymmetric profile with shoulders and new bands between 800 and 1300 cm^−1^ depending on the composition of the composite. For samples C and I, this profile has two shoulders centered at about ~1010 and 1090 cm^−1^ and the intensity of the band located at about 925 cm^−1^ was decreased. This evolution suggests a rearrangement of the host matrix, leading to the formation of chains or ring structures, namely, disilicate (Q3) and metasilicate (Q2) units.

By doping with lead content, the maximum of this profile shifts toward higher wavenumber values (~1110 cm^−1^) and the intensity of this band is stronger than its analogues. This feature corresponds to the Q4 unitswith four oxygen atoms per silicate of tetrahedrons overlapping with the stretching vibrations of Pb-O bonds in the [PbO_4_] structural units.

More significant differences are visible in the case of the addition of ash content in the host matrix when the presence of a new IR band centered at about 875 cm^−1^, including orthosilicate units was evidenced.

The observed features of the profile situated between 800 and 1200 cm^−1^ clearly indicate the progressive depolymerization of the silicate network by doping. Silicate tetrahedral units have different degrees of polymerization, namely, for doping with C and I, the formation of short ring structures, such as pyrosilicate and metasilicate units, by the addition of L amounts to the presence of tectosilicate units, and for ash content the orthosilicate units were evidenced. The effect of silicate network depolymerization increases by doping with lead and ash.

The prominent IR band centered at about ~3435 cm^−1^ is assigned to the H-O stretching vibrations from adsorbed water molecules on the surface of the sample and the presence of stretching vibrations of the Si-OH bonds. The IR band centered at about 1600 and 2900 cm^−1^ is assigned to the H-O-H bending vibrations and hydroxyl units [23]. For cast-iron (C) and iron (I) composite materials, these IR bands were gradually enhanced due to the presence of a higher number of Si-OH units.

The prominent IR band centered at about ~1450 cm^−1^ corresponds to the CO_3_^−2^ carbonate ions and the formation of CaCO_3_ crystalline phase in accordance with XRD data.

By doping of the glass network with different contents of wastes, structural modifications in the silicate network and in the process of water absorption were evidenced. The higher amounts of adsorbed water were evidenced in C and I samples, which have an important role in the hydration process of cement paste.

### 3.3. FTIR Spectra of the Composite-Cement Materials

In the Portland cement, the major anhydrous phases are alite, Ca_3_SiO_5_ (denoted as C_3_S) and belite, Ca_2_SiO_4_ (denoted as C_2_S), while the minor phases are calcium aluminate, Ca_3_Al_2_O_6_, calcite, CaCO_3_, gypsum, CaSO_4_, ferrite, Ca_2_(Al, Fe)_2_O_5_. By the reaction between cement and water, varied hydration products, namely, calcium silicate hydrates, C-S-H, portlandite, Ca(OH)_2_, ettringite, calcium monocarboaluminate, or calcium monosulphoaluminateare obtained [24]. The C-S-H consists of polymerized silica and calcium ions with water incorporated.

The formation of calcium silicate hydrate, C-S-H, occupies 50% ofthe fully hydrated cement paste. The C-S-H is a non-stoichiometric material with poor crystalline phase, which is the main hydration product and the most important component in the cement. The C_3_S and C_2_S are responsible for strength development performances in Portland cement from the short term up to months (C_3_S phase) and long term up to years (C_2_S phase).

The absorption intensities situated between 970 and 1100 cm^−1^ are due to the C-S-H calcium silicate hydrate. The dip in the IR bands situated between 800 and 970 cm^−1^ can be assigned to the dissolution of the C_3_S alite clinker phase, which can be correlated with the formation of C-S-H.

FTIR spectra of the validated and expired cement (EC) materials are plotted in Figure 3. The inspection of the IR bands located between 800 and 1100 cm^−1^ indicates the formation of C-S-H in the validated cement material due to the increase in the intensity of these bands.

The intensity of IR bands centered at about 1070 cm^−1^ and in the region between 1350 and 1550 cm^−1^ can be attributed to the CO_3_^−2^ carbonate ions and CaCO_3_, respectively. The intensities of these bands were increased for the expired cement.

A new IR band centered at about 855 cm^−1^ appears in the expired cement due to the Ca-O bond [25].

The IR band centered at about 3640 cm^−1^ corresponds to the Ca(OH)_2_, which is formed as a silicate phase in the dissolved cement [26]. In the validated cement, the intensity of this band was enriched suggesting the formation of C-S-H.

The FTIR spectra of the simple cement and composite-cement materials performed at 28 days after their preparation are displayed in Figure 4. For the expired cement and composite-expired cement materials, the IR spectra can be seen in Figure 5.

The IR bands characteristic of the sulphate originally from Portland cement are found in the range between 1100 and 1200 cm^−1^ due to the stretching vibration of S-O bonds in the SO_4_^−2^ units [9]. Ettringite is an hydrous calcium aluminum sulphate mineral and has a characteristic peak of SO_4_^−2^ vibrations centered at about 1115 cm^−1^ [27].

The strong IR band centered at about 1100 cm^−1^ corresponds to the antisymmetric stretching vibration of the Si-O-Si linkages, while the broad band situated at about 950 cm^−1^ reflects the stretching vibrations of the Si-OH [26,27].

Introduction of iron in the CI stone causes the increase in intensity of these IR bands. In all composite-expired cement materials, the characteristic feature of this band increases in strength and intensity.

For the expired cement, the intensity of the IR bands located between 800 and 1200 cm^−1^ was increased for all tested samples.

The use of C-S-H seeding demonstrates the improvement of the performance to strength since the microstructure of the hardened cement is fine and the hydration process was accelerated [28]. Introduction of iron content to the structure of validated cement causes the increase in the intensity of the IR band centered at about 970 and 3640 cm^−1^ (Figure 4), showing the formation of C-S-H and Ca(OH)_2_, respectively in the network structure. For the expired cement, the intensity of the IR band centered at about 970 cm^−1^ increases for all doped samples (Figure 5). The IR band corresponding to the Ca(OH)_2_ increases by doping, with the exception of adding lead content in the expired cement.

The characteristic feature of the IR band located at about 855 cm^−1^ was diminished for the expired cement and its intensity attains the maximum value for the samples ECA and ECC.

The analysis of IR data confirms that no new bands in the composite-cement materials appeared due to the addition of 2.5 weight% of composite product and the water associated bands was shifted slightly to lower wavenumbers.

### 3.4. NMR Relaxometry Investigations of the Composite-Cement Materials

The development in the microstructure of composite-cement materials during hydration was evaluated based on the surface area measured by ^1^H NMR relaxometry investigations. The echo trains measured by CPMG (Carr-Purcell-Meiboom-Gill) pulse sequence of the simple cement paste and composite-cement pastes performed at 28 days after their preparation are shown in Figure 6a. The full decay of the CPMG normalized echoes of the expired cement and composite-expired cement materials performed at 28 days after their preparation are shown in Figure 6b.

The possibility ofthe characterizationof water reservoirs and the evolution of pores inside the samples can be better identified using a numerical Laplace inversion for CPMG relaxation curves. The relaxation time distributions (T_2_) obtained by the inverse Laplace algorithm of the simple cement and composite-cement materials performed at 3, 7, 14, and 28 days after their preparation are presented in Figure 7.

In the relaxation time, distributions can be identified using four distinct water reservoirs inside the samples. The first peak (from the left) of the larger area can be assigned to the water bounded in the material components. The peaks situated at values of the higher relaxation time (from left to right) are attributed to the water in small pores, water in medium pores, and water in large pores [29]. For all investigated samples, the largest amounts of water are evidenced in the water bounded in the material components and few quantities of water correspond to the small, medium, and large pores. The small pores are responsible for the strong interactions of the cement material with liquid and gas water [30].

In the case of CI composite-cement materials containing iron powder, the evolution of the water peak in large pores was changed during 28 days after their preparation. The area of this peak remains constant up to 7 days, whereasit disappears suddenly for ≤14 days of hydration.

For the CC material, the water in the large pores is exhausted beyond 3 days. By adding lead powder in the CL paste, the water in the large pores was not evidenced after 3 days. Thereafter, a continuous increase arises in the intensity of this peak until the 28th day.

The peak of water in large pores disappears in the third day and after 28 days, whereas in the interval between the 7th and 14th day, the peak increased constantly for the CA material.

In all samples, the position of the largest peak classified as water bounded in the material components shifts to a smaller T_2_ relaxation time after 28 days of hydration comparatively with the simple cement paste.

After 28 days, the CI, CA, and CC composite-cement materials are not evidenced in water reservoirs of large pores. This evolution suggests that the water is consumed preferentially from the large pores during the hydration of the cement mixture. The position of the maximum of the main peak shifts to shorter T2 relaxation times for the CI and CA samples and moves to longer T2 relaxation times for the CC paste. The differences in the position of the water bounded in the cement materials can be attributed to the mobility (dynamics) of the material components [19]. The CL sample has less dynamic behavior.

The relaxation time distributions of the expired cement material and composite-expired cement materials performed at 3, 7, 14, and 28 days after their preparation are shown in Figure 8.

A small amount of pore water was present in the ECI sample after 14 days, and at 28 days, the peak corresponding to the water in the large pores was almost undetectable. The peaks assigned to the bounded water and water in the small and medium pores were increased continuously during the observed period. A trend of shifting these peaks toward higher relaxation times was also observed, indicating an increase in the mobility of the component of materials.

For the sample ECC, the major modifications in the shape and the intensity of the water peaks were evidenced after 14 and 28 days. At 14 days after the preparation, the peaks attributed to the bounded water and water in the small pores were overlapped. Up to 28 days, the bounded water increased again and a decrease in the signals due to the water in pores was clearly observed.

Up to 14 days, the peaks corresponding to the water in the small, medium, and large pores in the ECL sample continued to increase and an overlap of two peaks, namely, water in the small and medium pores, was clearly observed at 28 days.

The peak attributed to the water in small pores rises in signal and shifts to a larger T_2_ relaxation time, while the water reservoirs in medium and large pores were decreased abruptly until 28 days in the case of the ECA sample.

After 28 days, the chemically bound water clearly exhibited longer T_2_ relaxation time values than the expired cement sample.

In conclusion, the water reservoirs inside the composite-cement materials are mainly due to the consumption from large pores in the first stage. The process can be interpreted as the migration of water from large to small pores. Thereafter, with the acceleration of the hydration process, the water in medium and small pores can be quickly used for the hydration reaction. For the expired cement material, the water in large pores was evidenced in all samples up to 14 days. For the ECC and ECL samples at 28 days, the water was redistributed in the signal with a shorter relaxation time corresponding to the small and medium pores.

### 3.5. Vickers Hardness Measurements of the Composite-Cement Materials

To investigate the effect of waste powders on the Vickers hardness values of cement-based materials with recycled powder, ten different mixtures (five samples consisting of validated Portland cement and five samples used for the expired cement) were designed.

The influence of composites nature on Vickers hardness values distribution of composite-cement materials for the mixture with Portland cement and expired cement is shown in Figure 9.

In Figure 9a, it can be seen that the addition of L, A, C, and I composites in the validated cement material increases the Vickers hardness values. The maximum hardness values were attained for the CC and CI composite-cement materials. The IR data show that the C and I composites have a higher content of water in their structure. The water bounded to C and I components in the composite-cement materials lead to the difference in the mobility behavior, which increases the Vickers hardness values.

For the expired cement, the presence of ash-doped composite in cement materials yields a smaller hardness value than the undoped sample (Figure 9b). The rigidity of water bounded to material components and less mobility samples can be correlated with the smaller value of the Vickers hardness, in accordance with NMR investigations. A good correlation between the experimental hardness value and the mobility behavior of the components in the composite-cement materials predicted in NMR data was observed.

Briefly, the energy consumption and cost of nanocomposites preparation are lower than the cement production. The gas emissions of nanocomposite synthesis are significantly lower than the cement preparation, showing that the recycled nanocomposites are eco-friendly materials. The blending of nanocomposites in cement as supplementary cementing material and as a substitute of cement minimizes the consumption of cement and the global warming. The reprocessing of the waste into nanocomposites is beneficial for solving the problems of environmental pollution and for reducing the consumption of natural resources. Based on the results obtained, we can recommend the prepared composites as a replacement material for one part of Portland cement.

## 4. Conclusions

In this study, four composites were prepared and tested as a substituent of cement material. The structure of composite-cement materials was characterized by the analysis of IR and NMR spectra. The mechanical properties of these materials were determined by distributions of Vickers hardness.

XRD data evidence the presence of vitroceramic structure with diffraction peaks characteristic of the varied crystalline phases of the metallic ions. IR data show that the silicate network connectivity degree was decreased via doping. By doping of the expired cement, the IR band was enriched in all samples due to the formation of C-S-H. For the validated cement, the substitution of 2.5 weight% of iron produces an increase in this band. The formation of C-S-H is accompanied with a finer microstructure and performance of strength.

The four measured T2 peaks were evidenced in the NMR relaxometry, which consisted of four water reservoirs in the composite-cement materials. The Vickers hardness values were used as an indicator of the changes in the local mechanical properties of the composite-cement materials. The distribution and prediction models of Vickers hardness values were explored and correlated with the water bounded to the material components in the composite-cement materials. Based on the findings from this paper, the following conclusions can be noted: (i) The Vickers harness increases with the addition of 2.5 weight% metal composite (lead, iron, or cash iron powders) in the cement material, and (ii) for the ash-doped composite, the Vickers hardness value was lower at the expired cement.

## Figures and Tables

**Figure 1 materials-16-02398-f001:**
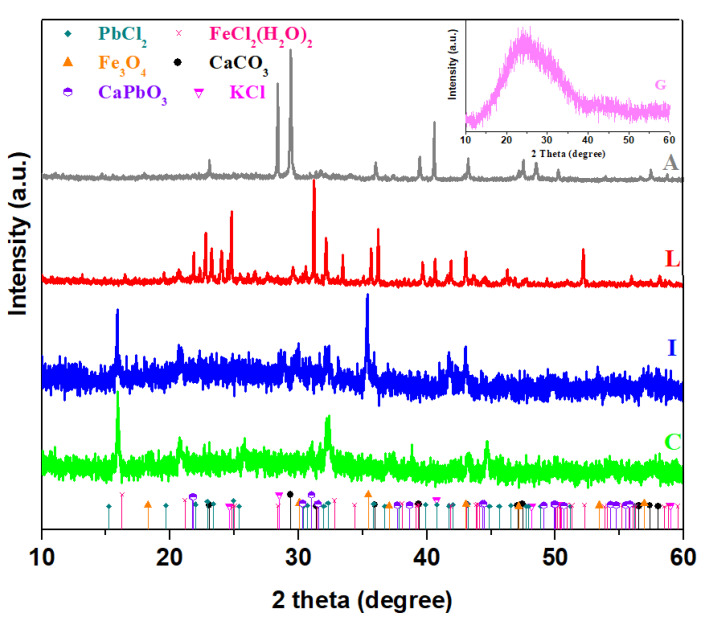
X-ray patterns of the broken glasses (G) and prepared composite materials (C—mixture of glasses and cash iron, I—mixture of glasses and iron powder, L—mixtureof glasses and lead powder, A—mixture of glasses and ash).

**Figure 2 materials-16-02398-f002:**
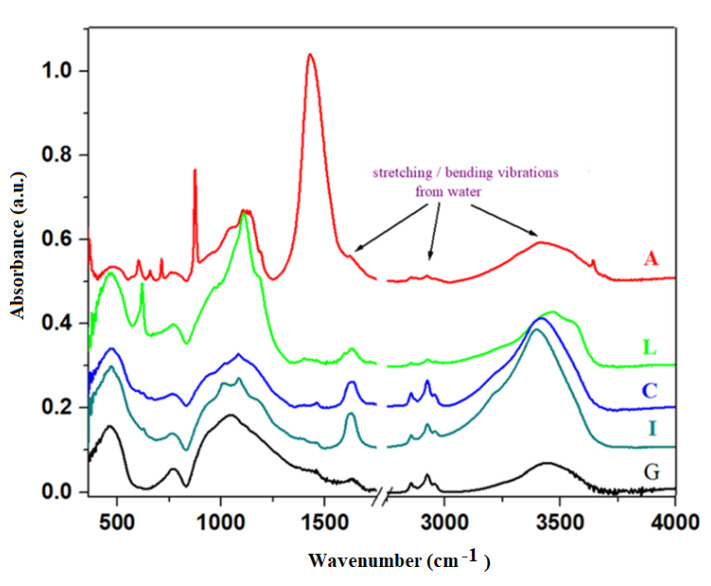
Fourier transform infrared (FTIR) spectra of waste glass powder and prepared composite materials.

**Figure 3 materials-16-02398-f003:**
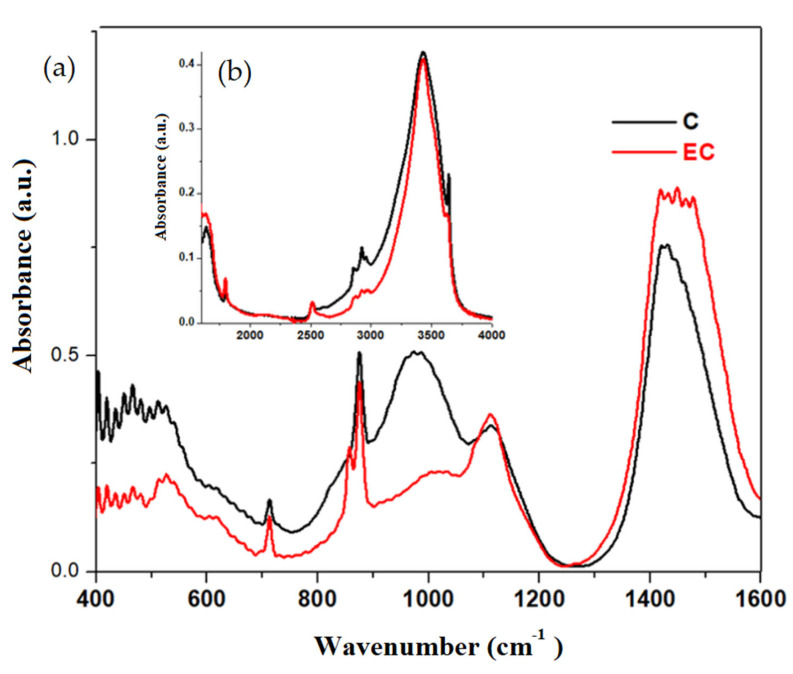
FTIR spectra of validated and expired cement materials performed at 28 days after their preparation in the region between (**a**) 400 and 1600 cm^−1^ and (**b**) 1600 and 4000 cm^−1^.

**Figure 4 materials-16-02398-f004:**
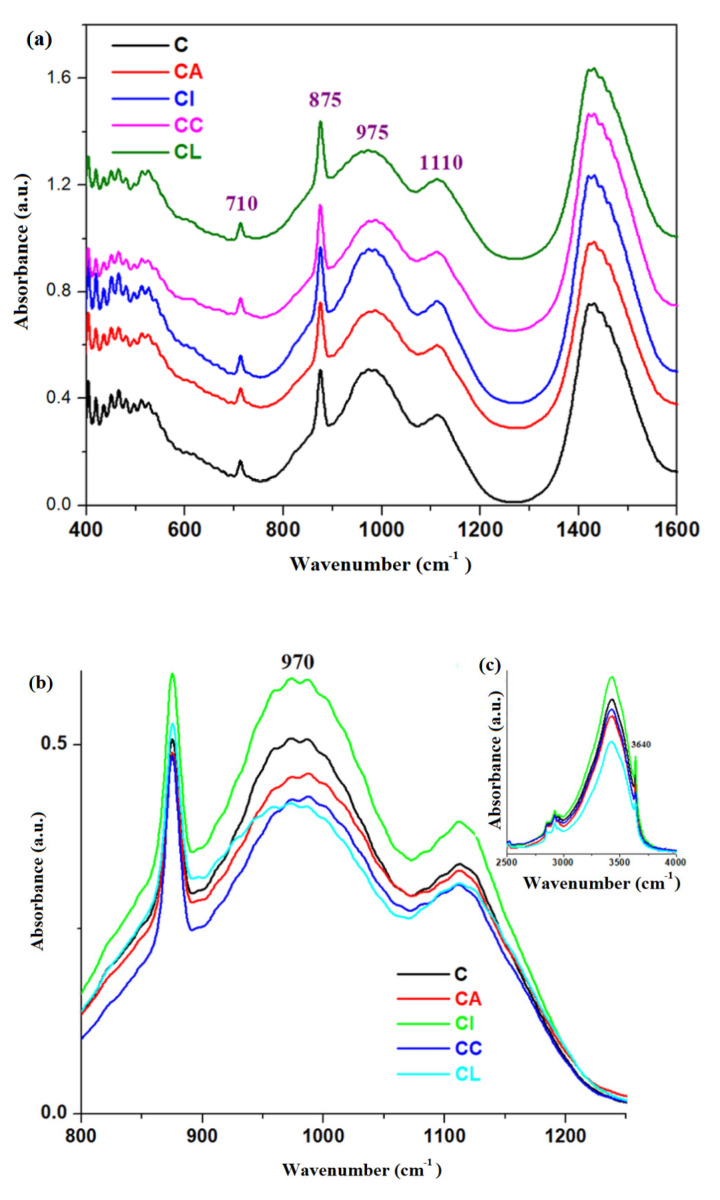
FTIR spectra of cement and composite-cement materials performed at 28 days after their preparation in the region between (**a**) 400 and 1600 cm^−1^, (**b**) 800 and 1250 cm^−1^,and (**c**) 2500 and 4000 cm^−1^.

**Figure 5 materials-16-02398-f005:**
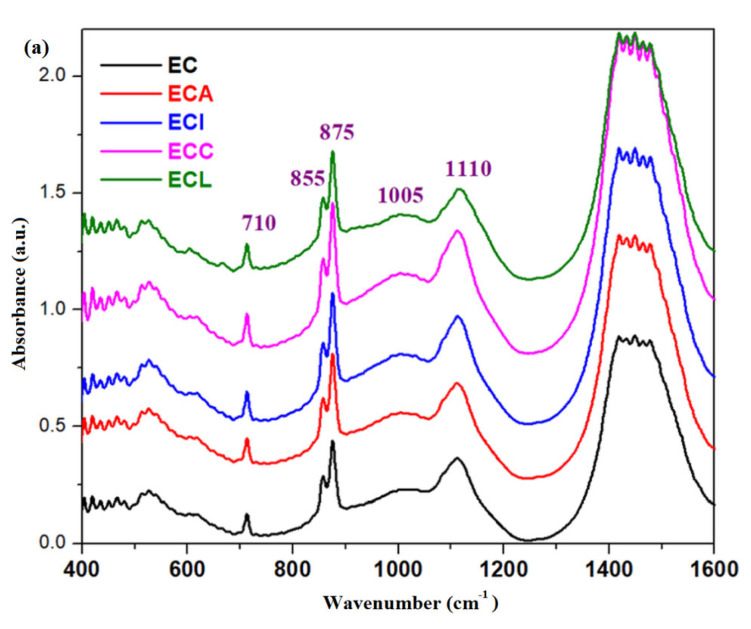
FTIR spectra of expired cement and composite-expired cement materials in the region between (**a**) 400 and 1600 cm^−1^, (**b**) 800 and 1200 cm^−1^,and (**c**) 2500 and 4000 cm^−1^ performed at 28 days after their preparation.

**Figure 6 materials-16-02398-f006:**
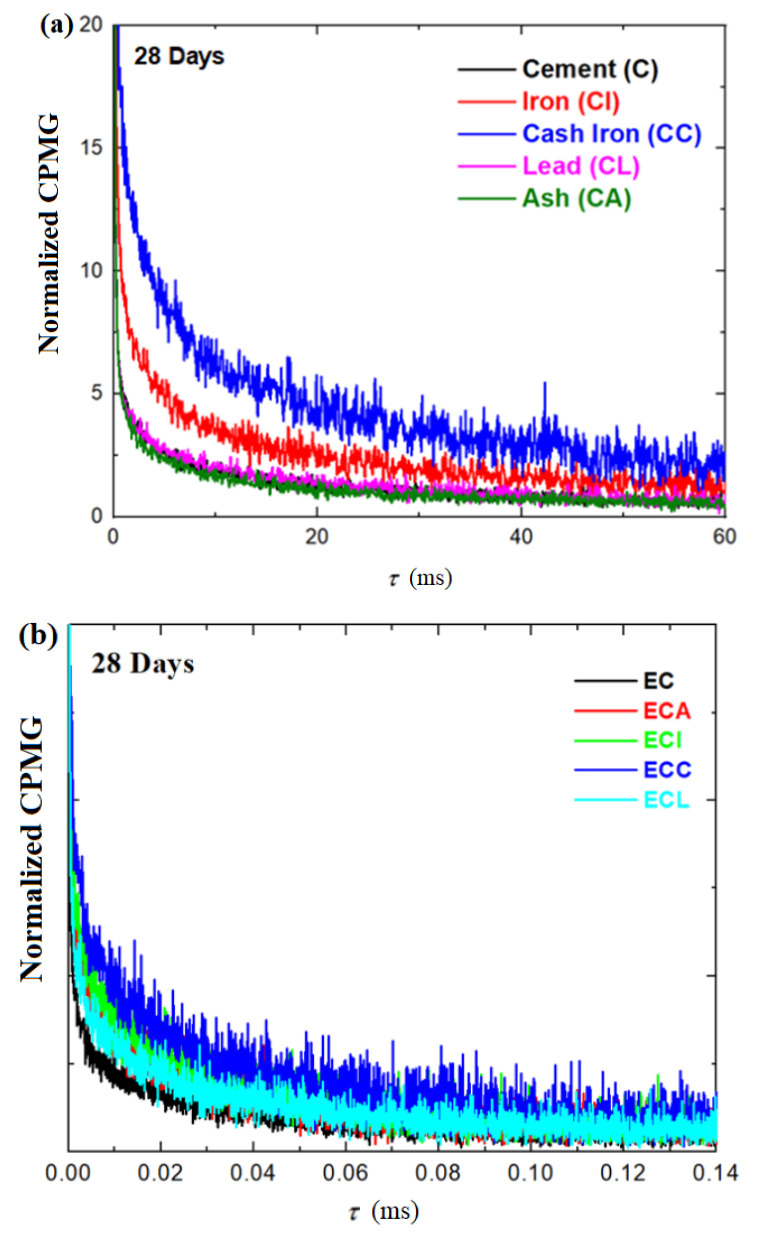
Normalized CPMG echo decay of (**a**) cement and composite-cement materials and (**b**) expired cement and composite-expired cement materials performed at 28 days after their preparation.

**Figure 7 materials-16-02398-f007:**
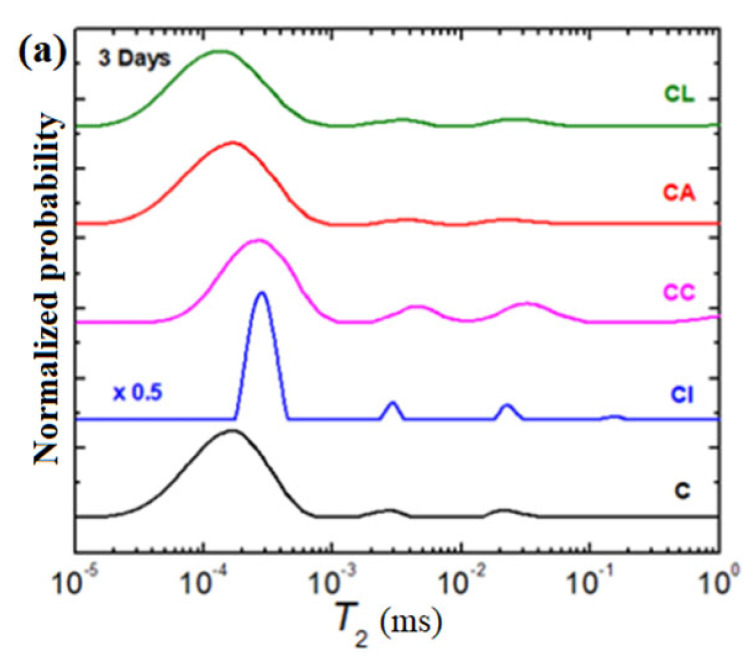
The relaxation time distributions of cement and composite-cement materials (CL: Lead composite-cement, CI: Iron composite-cement, CA: Ash composite-cement, CC: Cash iron composite-cement) performed at (**a**) 3 days, (**b**) 7 days, (**c**) 14 days, and (**d**) 28 days after their preparation.

**Figure 8 materials-16-02398-f008:**
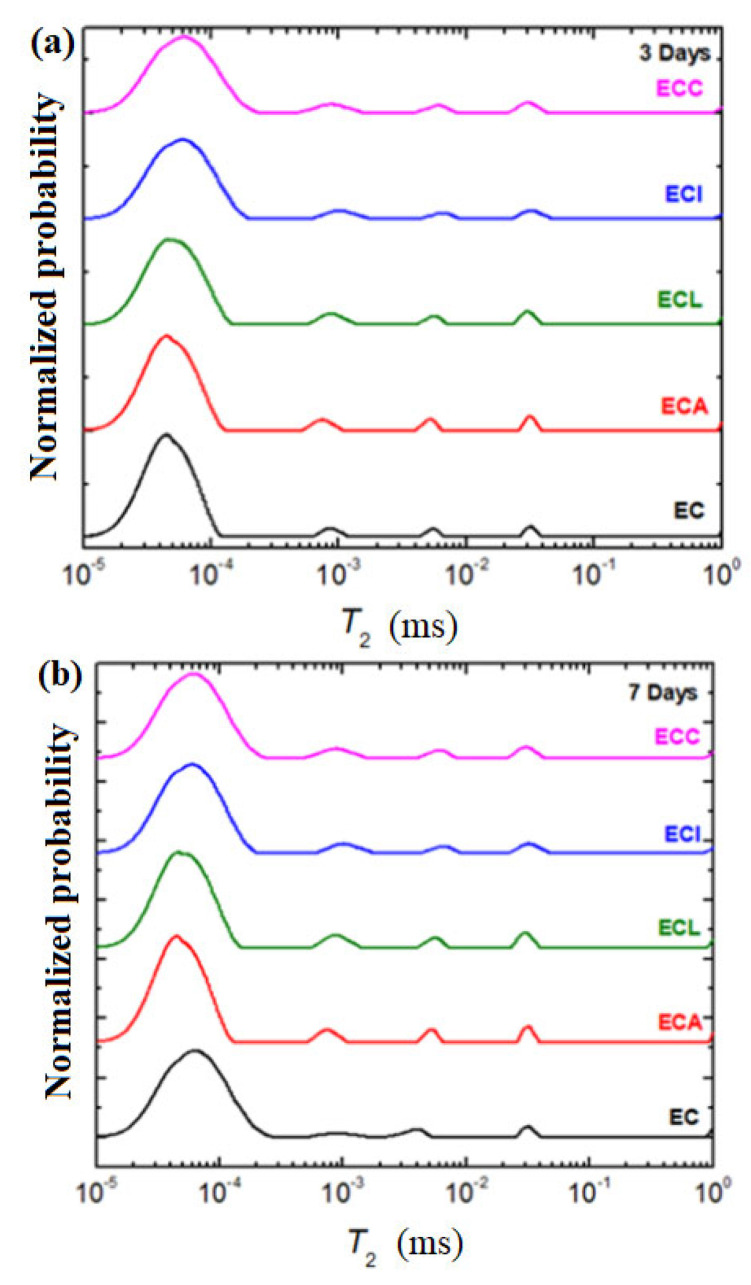
The relaxation time distributions of expired cement and composite-expired cement materials (ECL: Lead composite-expired cement, ECI: Iron composite-expired cement, ECA: Ash composite-expired cement, ECC: Cash iron composite-expired cement) at (**a**) 3 days, (**b**) 7 days, (**c**) 14 days, and (**d**) 28 days after their preparation.

**Figure 9 materials-16-02398-f009:**
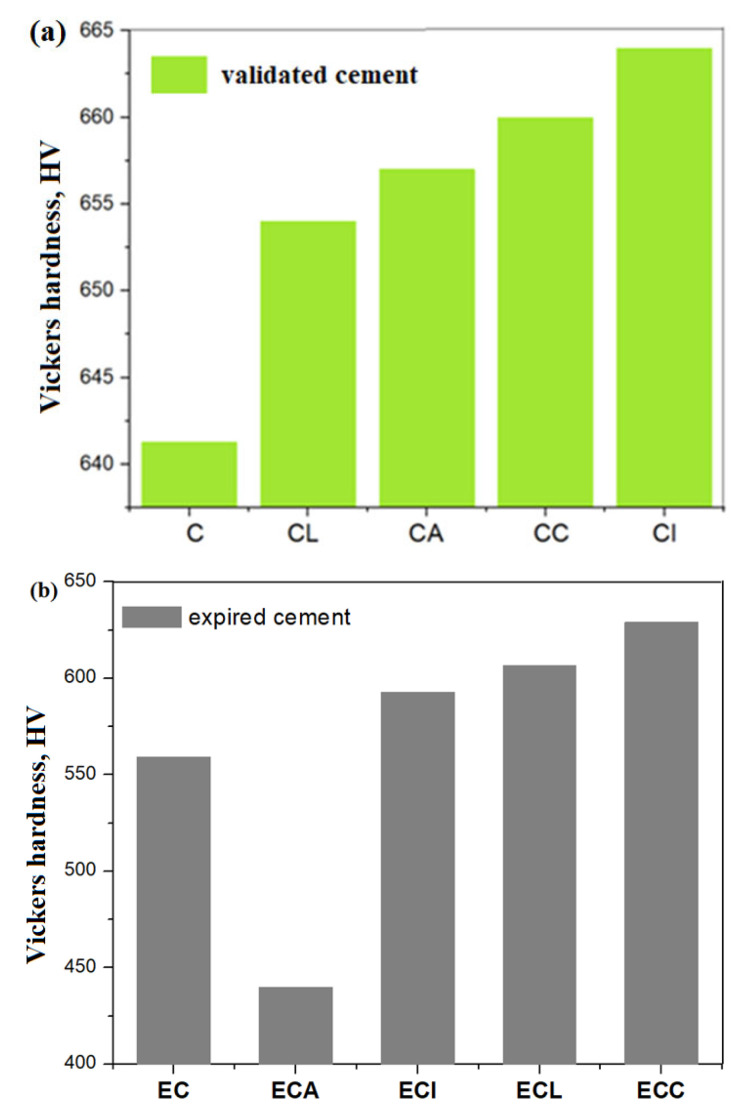
Influence of composites nature on the Vickers hardness values distribution of composite-cement materials for the mixture with (**a**) Portland cement and (**b**) expired cement, the measurements were applied after 28 days.

**Table 1 materials-16-02398-t001:** Description of composite and composite-cement materials.

Notation of Composite Material	Description of Composite Material	Notation of Composite-Validated Cement Materials	Description of Composite-Cement Materials	Notation of Composite-Expired Cement Materials	Description of Composite-Expired Cement Materials
I	Mixture of glasses and iron powder	CI	2.5 weight% of cement was substituted by I composite	ECI	2.5 weight% of cement was substituted by I composite
C	Mixture of glasses and cash iron powder	CC	2.5 weight% of cement was substituted by C composite	ECC	2.5 weight% of cement was substituted by C composite
L	Mixture of glasses and lead powder	CL	2.5 weight% of cement was substituted by L composite	ECL	2.5 weight% of cement was substituted by L composite
A	Mixture of glasses and ash	CA	2.5 weight% of cement was substituted by A composite	ECA	2.5 weight% of cement was substituted by A composite
G	Reference glass	C	Reference mix, 100% cement	EC	Reference mix, 100% expired cement

**Table 2 materials-16-02398-t002:** Average size of particle D of the high intensity peak of prepared composites.

Composite Material	Bragg Angle, θ (Degree) Corresponding to the Main Crystalline Phase	β (Radian × 10^−3^)	The Average Crystallite Size of Particle D (nm)
C (cash iron)	15.95 (FeCl_2_(H_2_O)_2_)	2.790	54.04
I (iron)	35.38 (Fe_3_O_4_)	2.792	63.59
	15.92 (FeCl_2_(H_2_O)_2_)	1.958	76.88
L (lead)	31.07 (CaPbO_3_) 24.77 (PbCl_2_)	1.919 1.958	88.07 81.43
A (ash)	29.39 (CaCO_3_) 28.38 (KCl)	2.199 1.759	75.56 93.55

## Data Availability

Not applicable.

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
