# Peer review of "Nanocomposites as Substituent of Cement: Structure and Mechanical Properties"

_materials, 2023, doi:10.3390/ma16062398_

Round 1

Reviewer 1 Report

Dear authors,

The manuscript, Title: Nanocomposites as Substituent of Cement: Structure and Mechanical Properties, presents new cement materials consist of 2.5 weight % composite originated from construction and demolition waste powder. The cement materials with recycled powder from construction and demolition wastes were characterized by: X-ray diffraction, FTIR and Nuclear Magnetic Resonance spectroscopy. The micro-Vickers hardness was measured to reflect the influence of composite nature in the local mechanical properties of the composite – cement for mix with Portland cement and expired cement.

The research, given that you deal with the recycling of waste material, should have its own technical and scientific value. Unfortunately, it is not clear why you decided for this combination of composite materials, nor for the percentage of composites (2.5%), as a replacement for Portland cement. I believe that with certain additions and good explanations, the manuscript can be published.

My questions and comments are in the manuscript (pdf file).

Sincerely

Author Response

The manuscript, Title: Nanocomposites as Substituent of Cement: Structure and Mechanical Properties, presents new cement materials consist of 2.5 weight % composite originated from construction and demolition waste powder. The cement materials with recycled powder from construction and demolition wastes were characterized by: X-ray diffraction, FTIR and Nuclear Magnetic Resonance spectroscopy. The micro-Vickers hardness was measured to reflect the influence of composite nature in the local mechanical properties of the composite – cement for mix with Portland cement and expired cement.

The research, given that you deal with the recycling of waste material, should have its own technical and scientific value. Unfortunately, it is not clear why you decided for this combination of composite materials, nor for the percentage of composites (2.5%), as a replacement for Portland cement. I believe that with certain additions and good explanations, the manuscript can be published.

My questions and comments are in the manuscript (pdf file).

Authors

In the Introduction section was detailed with the aim of this paper.

The authors thank the reviewer for their extremely useful comments.

Dr. Simona Rada

Reviewer 2 Report

I have read the Manuscript entitled "Nanocomposites as Substituent of Cement: Structure and Mechanical Properties" and found it very interesting.

In my opinion the work is comprehensive and attractive, however some minor modifications are required. My specific comments are as follows:

1. In the introduction, the author mainly discusses the recycling of the construction and demolition (C&D) wastes into secondary materials. But the main topic of this manuscript is about the structure and mechanical properties of nanocomposites. While the section presents a good overview of the topic, it lacks clarity in terms of the specific research question that the article is addressing. The author needs to clearly present the research problem and objectives of the study in the introduction.

2. There is not enough literature review and discussion of prior research that has been conducted in the field in the introduction. The authors could expand on the research gap that their study is aiming to address. As for pore structure and mechanical properties of cement, the author can refer to the related works to improve it, such as “Structure, fractality, mechanics and durability of calcium silicate hydrates (https://doi.org/10.3390/fractalfract5020047)”, “Pore structural and fractal analysis of the influence of fly ash and silica fume on the mechanical property and abrasion resistance of concrete (https://doi.org/10.1142/S0218348X2140003X)”, and “Fractal Analysis on Pore Structure and Modeling of Hydration of Magnesium Phosphate Cement Paste  (https://doi.org/10.3390/fractalfract6060337)” .

3. At the end of section 3.1, the author does not provide any explanation or discussion of the observed changes in particle size and structure upon doping with lead and ash.

4. In the second line of section 3.2, what is the "some specific domains" in FTIR analysis? This statement lacks clarity and precision, please define them. and the terminology used for the vibrational modes is inconsistent and sometimes incorrect.

5. Same as comment 3, section 3.2 does not provide any discussion or explanation of the observed changes in the FTIR spectra, for example, the enhanced IR bands for cash - iron (C) and iron (I) composite materials.

6. The results of the XRD and FTIR experiments and even some other contexts of the research are not clearly connected. The text makes no mention of how these findings of the characteristics of composite materials or their prospective could be applied in the cement industry. A more detailed discussion of the significance and implications of the results is needed.

Author Response

Dear Sir,

Concerning the above mentioned manuscript and related to the reviewer’s comments we state the following aspects:

Reviewer

I have read the Manuscript entitled "Nanocomposites as Substituent of Cement: Structure and Mechanical Properties" and found it very interesting.

In my opinion the work is comprehensive and attractive, however some minor modifications are required. My specific comments are as follows:

  1. In the introduction, the author mainly discusses the recycling of the construction and demolition (C&D) wastes into secondary materials. But the main topic of this manuscript is about the structure and mechanical properties of nanocomposites. While the section presents a good overview of the topic, it lacks clarity in terms of the specific research question that the article is addressing. The author needs to clearly present the research problem and objectives of the study in the introduction.

Authors

  1. The objects and research problem of this paper were added in the Introduction section.

Reviewer

  1. There is not enough literature review and discussion of prior research that has been conducted in the field in the introduction. The authors could expand on the research gap that their study is aiming to address. As for pore structure and mechanical properties of cement, the author can refer to the related works to improve it, such as “Structure, fractality, mechanics and durability of calcium silicate hydrates (https://doi.org/10.3390/fractalfract5020047)”, “Pore structural and fractal analysis of the influence of fly ash and silica fume on the mechanical property and abrasion resistance of concrete (https://doi.org/10.1142/S0218348X2140003X)”, and “Fractal Analysis on Pore Structure and Modeling of Hydration of Magnesium Phosphate Cement Paste  (https://doi.org/10.3390/fractalfract6060337)” .

Authors

  1. The Introdcution section was improved with literature review in the revised manuscript.

Reviewer

  1. 3. At the end of section 3.1, the author does not provide any explanation or discussion of the observed changes in particle size and structure upon doping with lead and ash.

Authors

  1. 3. The following sentences were added in the revised manuscript: “The literature data on particle size effect of recycled brick powder from blended cement paste show that the reduce of particles sizes improves the pore structure [16]. The higher particle sizes in the samples doped with lead and ash can have a significant impact on porosity and transport properties of cement structure at later ages.”

Reviewer

  1. In the second line of section 3.2, what is the "some specific domains" in FTIR analysis? This statement lacks clarity and precision, please define them. and the terminology used for the vibrational modes is inconsistent and sometimes incorrect.

Authors

  1. 4. The following phrases were added in the revised manuscript:

 “The analysis of IR data indicates specific domains of silicate units (800 – 1300 cm-1) and structural units of the metallic ions (370 and 550 cm-1). … The high intensity bands region located between 800 and 1300 cm-1 are assigned to the stretching vibrations of the Si - O bonds in varried silicate structural units. By doping this region differ from features of the host matrix because appears new shoulders and characteristic IR bands which indicates a drastic depolimerization of silicate network. These IR bands were assigned to symmetric Si - O stretching vibrations of silicate units containing [SiO4] tetrahedral units with zero, one, two, three and four non-bridging oxygens namely, tectosilicate, disilicate, metasilicate, pyrosilicate and orthosilicate units denoted as Q4, Q3, Q2, Q1 and Q0, respectively [19]. These IR features allows to distinguish bands located at about 1100-1250 cm-1, 1000-1100 cm-1, 900-1000 cm-1, near 900 cm-1, near 850 cm-1 each dominant at the tectosilicate, disilicate, metasilicate, pyrosilicate and orthosilicate units.

The “vibrational modes” were substituted by the “stretching vibrations”.

Reviewer

  1. Same as comment 3, section 3.2 does not provide any discussion or explanation of the observed changes in the FTIR spectra, for example, the enhanced IR bands for cash - iron (C) and iron (I) composite materials.
  2. The results of the XRD and FTIR experiments and even some other contexts of the research are not clearly connected. The text makes no mention of how these findings of the characteristics of composite materials or their prospective could be applied in the cement industry. A more detailed discussion of the significance and implications of the results is needed.

Authors

5, 6. A more detailed discussion of the XRD and FTIR data were presented in the revised manuscript. The applications of the composite materials were also evidenced.

The authors thank the reviewer for their extremely useful comments.

Dr. Simona Rada

Round 2

Reviewer 1 Report

Dear authors,

I am again sending materials-2244408-peer-review-v1-r1 pdf file containing my comments and questions.

Sincerely

Author Response

pdf file (materials-2244408-review_r2.pdf) contents the response for the reviewer
